# Genetic Diversity of the LTR Region of Polish SRLVs and Its Impact on the Transcriptional Activity of Viral Promoters

**DOI:** 10.3390/v15020302

**Published:** 2023-01-22

**Authors:** Monika Olech, Jacek Kuźmak

**Affiliations:** 1Department of Pathology, National Veterinary Research Institute, 24-100 Puławy, Poland; 2Department of Biochemistry, National Veterinary Research Institute, 24-100 Puławy, Poland

**Keywords:** small ruminant lentiviruses, LTR, transcription factors, promoter activity

## Abstract

A long terminal repeat (LTR) plays an indispensable role in small ruminant lentivirus (SRLV) gene expression. In this study, we present the LTR sequence of Polish SRLVs representing different subtypes, and analyzed their impact on SRLV promoter activity, as measured in transient transfection assays. Although certain nucleotide motifs (AML(vis), TATA box and the polyadenylation site (AATAAA)) were conserved across sequences, numerous mutations within the LTR sequences have been identified. Single nucleotide polymorphisms (SNPs) were detected in both regulatory (AP-1, AP-4, Stat and Gas) and non-regulatory sequences, and subtype-specific genetic diversity in the LTR region of Polish SRLVs was observed. In vitro assays demonstrated subtype-specific functional differences between the LTR regions of distinct SRLV subtypes. Our results revealed that the promoter activity of Polish strains was lower (1.64–10.8-fold) than that noted for the K1514 reference strain; however, the differences in most cases were not statistically significant. The lowest promoter activity was observed for strains representing subtype A5 (mean 69.067) while the highest promoter activity was observed for strain K1514 representing subtype A1 (mean 373.48). The mean LTR activities of strains representing subtypes A12, A17, A23, A18 and A24 were 91.22, 137.21, 178.41, 187.05 and 236.836, respectively. The results of the inter-subtype difference analysis showed that the promoter activity of strains belonging to subtype A5 was significantly lower than that for subtype A12 strains (1.32-fold; *p* < 0.00). The promoter activities of the A5 strain were 1.98-fold and 2.58-fold less active than that of the A17 and A23 strains, and the promoter activities of A12 strains were 1.955 and 1.5 times lower than the promoter activity of A23 and A17 strains, respectively. Furthermore, the promoter activity of A17 strains was 1.3 lower than the promoter activity of A23 strains. Our findings suggest that subtype-specific genetic diversity, mainly in the transcription factor’s binding sites, has an impact on their transcriptional activity, producing a distinct activity pattern for the subtypes. This study provides new information that is important for better understanding the function of the SRLV LTR. However, further research including more strains and subtypes as well as other cell lines is needed to confirm these findings.

## 1. Introduction

Small ruminant lentiviruses (SRLVs), which include the maedi-visna virus (MVV) and caprine arthritis encephalitis virus (CAEV), are a heterogeneous group of viruses belonging to the *Lentivirus* genus of the *Retroviridae* family, which infect sheep and goats globally. SRLVs are monocyte/macrophage tropic viruses, and viral replication is restricted until the maturation of monocytes to macrophages [1]. Transmission occurs mainly via the respiratory route and from mother to offsprings via the ingestion of infected colostrum/milk [2,3]. SRLVs can induce persistent systemic diseases affecting the lungs, mammary glands, synovial joints and central nervous system after long incubation periods. In the majority of cases, infection is asymptomatic. Only one-third of infected animals develop clinical manifestations such as interstitial pneumonia, mastitis, arthritis, dyspnea and, more rarely, encephalitis, ataxia or paralysis. There is no treatment for viral infection. Disease progression is usually slow, and the clinical outcome of infection is associated with the genetic background of the infected animal and the genome of the infecting strain [4,5,6,7].

The SRLV genome comprises two identical copies of single-stranded RNA (ss-RNA). This viral RNA is reverse transcribed into a double-stranded DNA, which is integrated into the host cell genome in the form of a provirus. The provirus is flanked by long terminal repeats (LTR) and contains a gene-encoding capsid (*gag*), envelope proteins (*env*) and enzymes (*pol*), as well as accessory proteins. The cells may either remain latently infected with little or no expression of the virus or be productively infected. In the latter cells, cellular transcription factors activate the transcription of the virus, which leads to viral gene expressions by interacting with specific DNA sequences. LTR is known to play a key role in transcriptional activation, virus integration and RNA polyadenylation. LTR consists of three regions, U3, R and U5, wherein U3 contains viral promoters and enhancer elements that interact with host proteins to form a complex network to regulate viral gene expression. In this region, different transcription-factor-binding sites (TFBS) have been described, such as AP-1, AP-4, E-box, AML (vis), gamma-activated sites (GAS), TNF-activated site (TAS) and IRF-1 [5,8,9,10].

SRLVs are characterized by high genetic variations leading to a variety of divergent strains and quasi-species. SRLVs are phylogenetically divided into five groups (A–E), which include different subtypes [11,12]. The majority of subtypes are able to cross the species barrier between sheep and goats under field conditions. To date, group A has 27 recognized subtypes (A1–A27), group B has 5 subtypes (B1–B5) and group E has 2 subtypes (E1 and E2) [11,13,14,15,16,17,18,19]. The transcription of the lentivirus genome depends on the presence of appropriate cellular transcription factors. Therefore, it is evident that sequence variations in the SRLVs LTR might affect the interaction with transcription factors and alter viral gene expression in infected cells, thus affecting virus-induced pathology and disease outcomes [10,20]. Several mutations, deletions or duplications in the LTR of SRLVs were shown to be associated with virulence, cellular tropism and pathogenesis [21,22,23,24,25]. Sequence variations were also observed between the different HIV-1 subtypes, and several subtype-specific sequence motifs in the HIV-1 LTR region have been identified [26,27,28]. Some subtype-specific variations have also been observed for SRLVs [29]. However, it remains unclear if this subtype-specific variability in the LTR region could influence viral transcription efficiencies. In this study, we present the LTR sequence of Polish SRLVs representing different subtypes and analyzed their impact on SRLVs promoter activity as measured in transient transfection assays.

## 2. Materials and Methods

### 2.1. Sample Collection and DNA Extraction

The 22 blood samples analyzed in this study originated from sheep and goats that were naturally infected with an SRLVs from 5 herds within different regions in Poland. All samples were tested previously and were PCR-positive for an SRLVs [11,30]. Sample animals were clinically healthy, without any clinical signs of SRLVs. Genomic DNA was extracted using a Dneasy Blood & Tissue Kit (Qiagen, Valencia, CA, USA) following the manufacturer’s instructions, and the samples were stored at −20 °C until use. DNA from the K1514 reference strain (pBSCA-K1514) was a generous gift from Dr. Yahia Chebloune, PAVAL Laboratory, France.

### 2.2. Amplification and Construction of LTR Plasmids

To perform a transcriptional analysis comparing the LTRs from different SRLVs variants, the LTR fragment was amplified using the standard PCR protocol with primers LTREFW (5′GCGCTCGAGACTGTCAGGRCAGAGAACARATGCC3′; *XhoI* site underlined) and LTRERW (5′ GCGAAGCTTCTCTCTTACCTTACTTCAGG3′; *HindIII* site underlined), as described by Ryan et al. [1], including the *XhoI* and *HindIII* restriction sites. PCR was performed in reaction mix volumes of 25 µL. The mixture contained 500 ng of genomic DNA, 2.5 µL of buffer (10×), 0.5 µL of dNTP solution (10 µM), 1 µ Mof gCl_2_ (25 mM), 0.125 µL of DreamTaq polymerase (200 U) and 0.75 µL of each primer (10 µM). The thermal conditions were as follows: 1 min at 94 °C, 35 cycles (30 s at 94 °C, 30 s at 55 °C, 30 s at 72 °C) and the final extension step at 72 °C for 10 min. PCR products were separated and analyzed by electrophoresis on 1.5% agarose gel containing SimplySafe (EURx, Gdańsk, Poland) and diluted at 1:10,000 in 1× TAE buffer. The amplicons representing LTR sequences were purified on NucleoSpin Gel and PCR Clean-up columns (Macherey-Nagel GmbH & Co KG, Dueren, Germany) and cloned into a pDrive vector using a QIAGEN PCR Cloning Kit, according to the manufacturer’s protocol. Ligation products were used to transform EZ Competent Cells (Qiagen), and plasmid DNA was extracted using a NucleoSpin Plasmid kit (Marcherey-Nagel GmbH & Co KG, Germany). Next, the LTR inserts were sub-cloned in the *XhoI* and *HindIII* restriction sites of pGL4.10 vector (Promega, Madison, WI, USA) after *XhoI/HindIII* digestion. Chemically competent Escherichia coli One Shot TOP10 cells were used for the transformations and maintenance of the plasmid DNA. Finally, bacterial cultures for the preparation of the LTR plasmids were grown for 14 h in a Luria Bertani (LB) medium supplemented with ampicillin (100 µg/mL) and purified using a NucleoSpin Plasmid Kit according to the manufacturer’s protocol. The obtained recombinant plasmids containing different LTR inserts were confirmed by sequencing using a Big Dye Terminator v3.1 Cycle Sequencing kit on a 3730xl DNA Analyzer (Applied Biosystems, Foster City, CA, USA). The direct sequencing of PCR products was also performed to check if recombinant plasmids constitute the most abundantly represented sequences. The generated plasmids were used to transfect immortalized caprine fibroblast cells (TIGEFs). The cells were maintained in an RPMI 1640 medium (Gibco, Thermo Fisher, Waltham, MA, USA) supplemented with 5% heat-inactivated fetal calf serum (Gibco, Thermo Fisher, USA) and 1% penicillin–streptomycin mixture (Lonza BioWhittaker, Verviers, Belgium), and they were cultured at 37 °C in an incubator with 5% CO_2_.

### 2.3. Transfection and Luciferase Reporter Gene Assay

One day before transfection, the T-immortalized goat embryonic fibroblast (TIGEF) cells were seeded in 6-well plates at a density of 1 × 10^5^ cells per well (Nunc, Thermo Fisher Scientific, Eindhoven, The Netherlands). The cells were transfected with 200 ng of test plasmids using 1.6 µL of the FuGENE^®^ HD Transfection Reagent (Roche). The pGL4.13 plasmid (Promega, USA), expressing the luc2 reporter gene under the control of the SV40 early enhancer/promoter, was used as a positive control, and the empty vector pGL4.10 as a negative control. At the same time, 20 ng of pGL4.73 (Rluc/SV40) Vector (Promega, USA) was co-transfected, so firefly activities were standardized according to Renilla luciferase activities. Then, 24 h after transfection, the cells were washed with phosphate-buffered saline (PBS), and cell extracts were prepared with a Passive Lysis Buffer (Promega, Madison, WI, USA). The firefly and Renilla luciferase activities were measured using a Dual-Luciferase^®^ Reporter Assay System (Promega, USA) following the manufacturer’s instructions and a Centro XS3 LB 960 Microplate Luminometer (Berthold Technologies, Bad Wildbad, Germany). All experiments were performed in triplicate. The firefly luciferase activity was normalized to Renilla luciferase activity (Firefly Luciferase/Renilla Luciferase) and presented as relative luciferase activity.

### 2.4. Phylogenetic Analysis

The LTR sequences were analyzed using MEGA 6 software [31]. Multiple sequence alignments were performed using multiple sequence comparisons by applying the log-expectation (MUSCLE) algorithm. Model testing was performed to select the best-fitted evolutionary model based on the Bayesian information criterion (BIC) and Akaike information criterion (AIC). According to the results, the Kimura 2-parameter (K2) model with the gamma distribution (+G) and 5 rate categories was used to construct a phylogenetic tree using neighbor-joining (NJ) methods. A bootstrap (1000 iterations) analysis was performed to test the statistical robustness of the tree. The prediction of potential transcription factor binding sites (TFBS) was performed using the Jaspar algorithm [32] with a relative profile score threshold of 80% or manually. The sequences were submitted to the Gen-Bank database under the following accession numbers: OP901447–OP901471.

### 2.5. Statistical Analysis

All statistical analyses were performed using STATISTICA version 10 (StatSoft, Tulsa, OK, USA), where a *p* value of <0.05 was considered significant. Differences in the luciferase activities between the two groups were analyzed by the Mann–Whitney U Test.

## 3. Results

### 3.1. Phylogenetic Analysis

The LTR sequences obtained from the PCR products and recombinant plasmids were identical, indicating that the cloned LTRs reflect the most abundantly represented viral sequences. Based on the multiple sequence alignment of 22 sequences of Polish strains analyzed in this study and reference sequences, a phylogenetic tree was constructed using the neighbor-joining method (Figure 1). The mean genetic distance of the LTR sequences analyzed in this study was 14.2% and ranged from 0.5% to 24.3%. Eight sequences (g7096, g7134, g7102, g6808, s20, g8891, g3533 and g9509) were closely related, with strains belonging to subtype A12 (mean nt genetic distance 4.6% ± 0.7%). One ovine sequence (s1) was closely related to sequences belonging to subtype A24 (mean nt genetic distance 1.0% ± 0.4%), while two sequences (g1561 and g5616) originating from goats were closely related to A17 strains (mean nt genetic distance 4.4% ± 0.9%). Eight sequences from goats (g8008, g5994, g5819, g4742, g5870, g6038, g5962 and g3038) were clustered within subtype A5 (mean nt genetic distance 2.1% ± 1.4%). One sequence (g4464) from a goat was closely related to sequences belonging to subtype A18 (mean nt genetic distance 4.1% ± 0.9%), while two ovine sequences (s4018 and s3275) were closely related to strains representing subtype A23 (mean nt genetic distance 5.7% ± 1.1%).

### 3.2. Analysis of Promoter Sequences

Multiple motifs were identified within the LTR sequences, including AP-1 sites, AML, AP-4, TATA box, poly A, CAAAT, GAS, TAS, Stat-1, NF-κB and IRF-1. Numerous mutations within the viral promoter have been identified in field isolates. AML (vis), TATA box and the polyadenylation site (AATAAA) were the most conserved elements between all analyzed strains. Our results revealed that strains representing subtype A17 had a unique T to A substitution in the fifth position of TATA box. AP-4 was also well conserved, showing only minor variabilities between sequences belonging to different subtypes. Six variably degenerate AP-1 sites were observed in sequences analyzed in this study. The AP-1 sites revealed rather subtype-specific conservation. Three Stat-1 sites and one GAS region were identified. Some point mutations were observed in both regions. Our results revealed that only the A5 strains had three Stat-1 sites, while in the other analyzed strains, only two Stat-1 sites were identified. Furthermore, sequences belonging to subtype A5 had two NF-κB sites that were not present in the sequences of the other subtypes. The IRF-1 site was also identified in 8 out of 23 sequences, which belonged to subtypes A12 (7 strains) and A23 (1 strain). Furthermore, a sequence from subtype A1 had two unique insertions in the U3 region: one (10 nt GGTCATGTCA) located near the 5′ end of the U3 region, and the second (21 nt GGATGACACAGCAAATGTAAC) located in the central region of the U3. The latter contains additional putative AP-1 and AML (vis) binding sites and the CAAAT sequence. No transactivation response elements were identified in the R region. However, strains representing subtypes A1 and A17 had a specific 11 nt (CGAAGGAAAGA/G) insertion in the R region. Our results showed that the LTR sequences reflected subtype-specific patterns of sequence diversity (Figure 2). The number of repeats of the CAAAT sequence was different depending on the strains. Strains K1514 and 1561 (A17) had three copies of the CAAAT sequence, while strains 3533 (A12), 4742 (A5), 5962 (A5), 3038 (A5), 8008 (A5), 4464 (A18) and 5616 (A17) had two copies of CAAAT. Strains 6038 (A5), 5819 (A5), 5994 (A5), 5870 (A5), 1 (A24), 20 (A12), 9509 (A12), 6808 (A12) and 7096 (A12) had only one copy of CAAAT, and strains 7134 (A12), 7102 (A12), 8891 (A12), 4018 (A23) and 3275 (A23) had no copies.

### 3.3. Analysis of LTR Promoter Activity

To evaluate the effect of the basal transcriptional activity of Polish field SRLV strains representing different subtypes, the LTR sequences of these strains were subcloned into the pGL4.10 vector, a luciferase-based reporter. At 24 h post-transfection, the transcriptional activity of the LTR promoter of tested strains was examined by the luciferase gene activity in TIGEF cells. The promoter activity measured by luciferase expression is shown in Figure 3. Different levels of luciferase activity were observed between the strains. The results of the analysis showed that the transcriptional activity of Polish SRLV strains was higher (3.2–21.9-fold) compared to pGL4.13-control plasmids containing the SV40 promoter. The strongest enhancer activity was achieved with reference strain K1514 and was 34.5 times higher than that of pGL4.13.

Our results revealed that the promoter activity of Polish strains was lower (1.64–10.8-fold) than that noted for reference strain K1514; however, the differences in most cases were not statistically significant. Only the promoter activity of A5 strain 6038 was significantly lower than that of strain K1514 (*p* > 0.035). The lowest promoter activity was observed for strains representing subtype A5 (mean 69.067), while the highest promoter activity was observed for reference strain K1514, representing subtype A1 (mean 373.48). The mean LTR activities of strains representing subtypes A12, A17, A23, A18 and A24 were 91.22, 137.21, 178.41, 187.05 and 236.836, respectively (Figure 4).

The results of the inter-subtype difference analysis showed that the promoter activity of the strains belonging to subtype A5 was significantly lower than that for subtype A12 (1.32-fold; *p* < 0.00) (Figure 4) strains. Differences between the promoter activity of subtypes A1, A17, A18, A23 and A24 were also noted; however, these subtypes were not included in the inter-subtype analysis due to there being limited samples (<5) representing each subtype. Nevertheless, the comparison of subtypes comprising at least two strains showed that the promoter activities of the A5 strains were 1.98-fold and 2.58-fold less active than that of the A17 and A23 strains. The promoter activities of the A12 strains were 1.955 and 1.5 times lower than the promoter activity of the A23 and A17 strains, respectively. Furthermore, the promoter activity of the A17 strains was 1.3 times lower than the promoter activity of the A23 strains.

## 4. Discussion

The expression of proviral DNA is favored by transcriptional activation in the LTR, which is regulated by transcription factor’s binding sites within the U3 region. Several putative TFBSs, including the AML (vis), TATA box and polyA sites, were highly conserved among the SRLV LTR sequences, as shown in the sequences obtained in our study. Although certain nucleotide motifs were conserved across the sequences of the Polish strains, numerous mutations within the LTR sequences were identified. Single nucleotide polymorphisms (SNPs) were detected in both the regulatory (AP-1, AP-4, Stat and Gas) and non-regulatory sequences of the U3, R and U5 subregions. The detection of SNPs in regulatory elements suggests that there may be differential SRLV expressions in hosts infected with a particular subtype. Sequence variations were also observed between different HIV-1 and HIV-2 subtypes, and it was revealed that the subtype-specific LTRs dictate different promoter activity and replication rates, which resulted in biological differences between the subtypes [26,27,28]. Thus, we suppose that the sequence variation between the SRLV subtypes may also correspond to the different transcriptional activities of these strains. To test this, we cloned the LTR sequences of Polish field SRLVs representing different subtypes and investigated their in vitro transcriptional activities.

All subtype LTRs were found to be functional promoters. Different levels of promoter activity were observed for strains representing different subtypes. The lowest promoter activity was observed for strains representing subtype A5, while the highest promoter activity was observed for the K1514 reference strain representing subtype A1. Our results revealed that the LTR from the K1514 strain contained 10 nt and 21 nt insertions in the U3 region, which were not present in the sequences of the other analyzed strains. While no putative transcription factor binding domains were identified within the 10 nt insertion, a 21 nt repeat contains potential transcription factor binding sites for AP-1, AML(vis) and the CAAAT sequence. The AML (vis) motif is a recognition site for the members of the AML/PEBP2/CBF transcription factor family. Polyomavirus enhancer-binding protein 2 (PEBP2) and core-binding factor (CBF) were identified as proteins that regulate the transcription of viral and cellular genes [20]. The AML (vis) sites were highly conserved among all analyzed sequences, and it seems that the number of AML (vis) sites present in the LTR plays an important role in the regulation of SRLV replication. This notion is supported by the findings that MVV-AML-defective chimeric viruses showed both lower replication in SPC cells and lower promoter activity [20,23]. No transcriptional activity was observed when the AML site (vis) was absent [33,34]. Furthermore, it is also expected that the CAAAT sequence that constitutes an E-box may influence viral transcriptions in SCP cells. Poor viral growth occurs when a single CAAAT sequence is present, and no transcription occurs when CAAAT is deleted [22]. Our study revealed that the K1514 strain that showed the highest promoter activity had the highest number of CAAAT copies. However, the correlation between the number of copies of CAAAT and the level of promoter activity was not observed when other strains were analyzed. The finding that both the CAAT sequence and the AML site, when present in one copy, contribute to the transcription efficiency in SCP cells supports the hypothesis that transcription is synergistically controlled and that only a repeated sequence containing both sites creates a redundancy in terms of transcription factor binding sites [22]. The K1514 strain also possesses additional sequences of the AP-1 site, which are important for the regulation of SRLV expression. Six variably degenerate AP-1 sites were observed in the sequences analyzed in this study. All Polish strains had five AP-1 sites that showed subtype-specific patterns. Differences in AP-1 sequences have also been observed by other authors [35,36,37]. Furthermore, Sutton et al. [33] revealed that substitutions in the AP-1 sites do not rule out a low affinity for transcription factors and that they can continue to have basal promoter activity, although alterations in the cis-regulatory function have been observed. Thus, we hypothesize that variations in these transcription factor binding sites may reflect differences in the transcriptional activity of strains belonging to different subtypes; however, further studies should be conducted to confirm this assumption.

The detailed characterization of the single binding sites performed by Juganaru et al. [38] revealed that only AP-4 sites are crucial or even essential for enhanced LTR promoter activity. Our study revealed that the AP-4 site was quite conserved with few subtype-specific variations. Specifically, all A5 strains had the unique substitutions of T to G and T to C in the first and ninth positions of the AP-4 site, respectively, which may also impact the different promoter activity of these strains. The TATA box located in the U3 region is an important element as a transcription promoter, and the deletion of the TATA box abolished transcription almost completely [22]. Our results revealed that strains representing subtype A17 had a unique substitution of T to A in the fifth position of the TATA box. The same substitution was observed in SRLV strains belonging to subtypes A27, A20 and B3 and strains EV-1 and SAOMVV [29,35]. Within the group of retroviruses, there are several species that also have the same mutation in the TATA box. While all HIV-1 clades show the TATAA box, the J and E subtypes contain a TAAAA sequence [27]. Furthermore, several types of simian immunodeficiency virus (SIV) also possess the TAAAA sequence [39]. The significance of this mutation is still unknown. Jeeninga et al. [26] revealed that a mutation in the TATA box did not affect the promoter activity of the E LTR subtype of HIV-1, indicating that the TAAAA motif functions efficiently despite the mutation in this box. On the one hand, in vitro and in vivo transcription analyses performed by van Opijnen et al. indicated that a mutation in the TATA box dramatically decreases HIV gene expression [39].

Our results also revealed that the sequence derived from the K1514 strain and two strains representing subtype A17 had an 11 nt deletion in the R region. This deletion was previously described by Angelopoulou et al. [21], who suggested that the presence of this specific deletion may be associated with the lower pathogenicity of SRLV strains. The R region does not harbor elements that participate in the control of transcription. In HIV, the Tat-activating region’s (TAR) motif is encoded by the transcribed R region of LTR sequences; thus, the RNA secondary structure of this motif appears to be critical for the action of the Tat regulatory protein of these viruses, which functions to increase viral gene expression [27]. The role of the R region in the transcriptional regulation of SRLV is unknown. The basal activity of the SRLV promoter does not require the trans-activating protein Tat. This may be why the R region in the LTR sequences of SRLV is quite variable.

SRLV transcription is thought to be partially regulated and dependent on the presence of pro-inflammatory cytokines. It was demonstrated that interferon gamma (IFN-γ) and tumor necrosis factor alpha (TNF-α) activate viral transcriptions via the GAS and TAS sites, respectively [40]. The mentioned cytokines activate the viral promoter in the U3 region and is mediated via the JAK-STAT pathway [40,41]. These sites and the GAS and TAS response elements were identified in the U3 region of the sequences analyzed in this study. While no subtype-specific mutations were detected in the GAS region, such subtype-specific SNPs were identified in the TAS region. The substitution of T to C was characteristic for the A12 strains, while the substitution of T to G was characteristic for the A17 and A23 strains. Furthermore, the substitution of A for C in the TAS region was only observed in A5 strains. It is known that minor variances within the transcription factor binding sites in the LTR of retroviruses can have a significant impact on their activity [42]. Therefore, we suppose that the identified point mutations may also have an impact on the promoter activity of the Polish strains that produce a distinct activity pattern for the subtypes.

Our study revealed that subtype A5 strains that had the lowest promoter activity had the most distinct promoter architecture. Only the LTR sequences of the A5 subtype strains contained two binding sites for nuclear factor kappa B (NF-κB). The number of NF-κB sites differed among the HIV-1 subtypes. While subtype B had two adjacent NF-κB sites, three NF-κB sites are present in subtype C, and the CRF01_AE HIV-1 LTR contains a single NF-κB site [26]. Furthermore, Jeeninga et al. [26] showed a correlation between the number of NF-κB sites and the level of TNF-α stimulation. Furthermore, it was revealed that the number of NF-κB binding sites influenced the subtype-specific HIV-1 LTR activity, as the depletion or insertion of one copy of this site significantly altered the LTR activity on TNF-α stimulation [42]. IFNγ also induces the transcriptional regulation of target genes via the recognition of promoter consensus sequences by transcription factor Stat-1. The production of IFNs and the activation of Stat1 represent key steps in the immune response against viruses. Stat1 activates the transcription of genes with crucial antiviral properties [43]. Our results revealed that only the A5 strains had three Stat-1 sites, while in the other analyzed strains, only two Stat-1 sites were identified; one was more conserved, and the second was more variable. However, subtype-specific variations were observed in both, especially for A5 strains. The role of the STAT1 pathway in the regulation of promoter activity is unknown and warrants further studies.

LTR-mediated gene expression may be modulated by different transcription factors in a cell-type-dependent manner. The cell type and its differentiation state may lead to variations in the transcriptional activity of an LTR. Here, we reported measurements of LTR-driven gene expression upon the transfection of TIGEF cells. Therefore, the presence of a cell-type specific combination of transcription factors in TIGEF cells may generate distinct transcriptional activity in comparison to other cell lines.

In conclusion, our results provide further insight into subtype-specific differences in the SRLV LTR. Our findings suggest that subtype-specific genetic diversity, mainly in the transcription factor binding sites, impacts their transcriptional activities, producing a distinct activity pattern for the subtypes. This study provides new information that is important for better understanding the function of the SRLV LTR. However, further research, including more strains and subtypes as well as other cell lines, is needed to confirm these findings. It is also important to perform assays with mutant strains to discern the functionality of the identified subtype-specific sites.

## Figures and Tables

**Figure 1 viruses-15-00302-f001:**
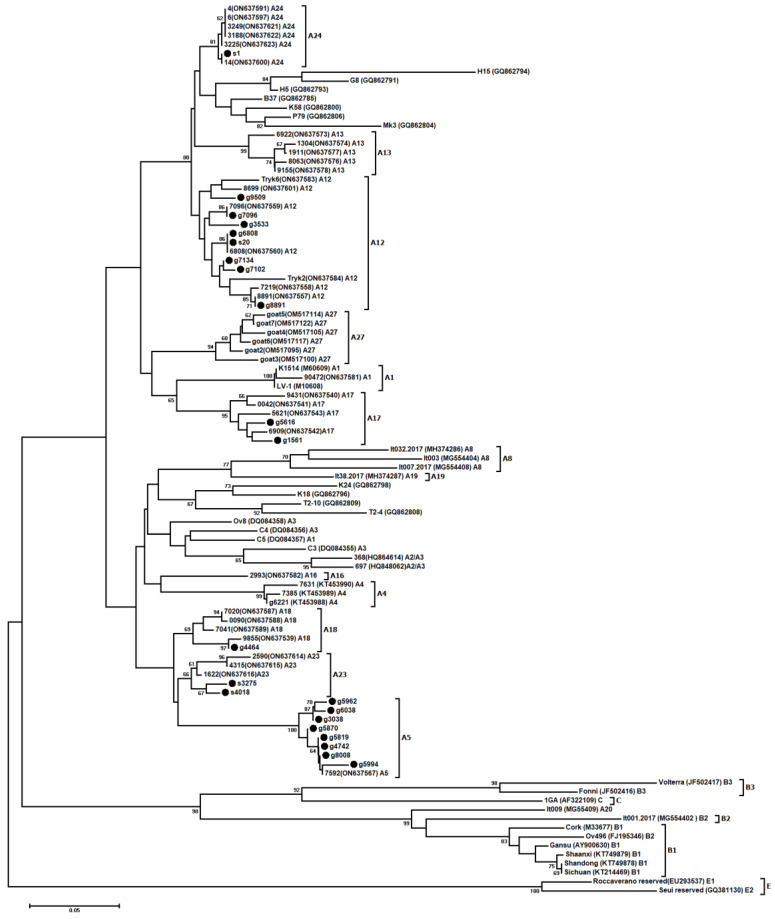
Neighbor-joining phylogenetic tree based on the alignment of the LTR fragment. Sequences from this study are labeled with black circles (s—sheep; g—goat). Reference SRLV strains are shown by name followed by their GenBank accession number and subtype. Numbers at the branches indicate the percentage of bootstrap values obtained from 1000 replicates.

**Figure 2 viruses-15-00302-f002:**
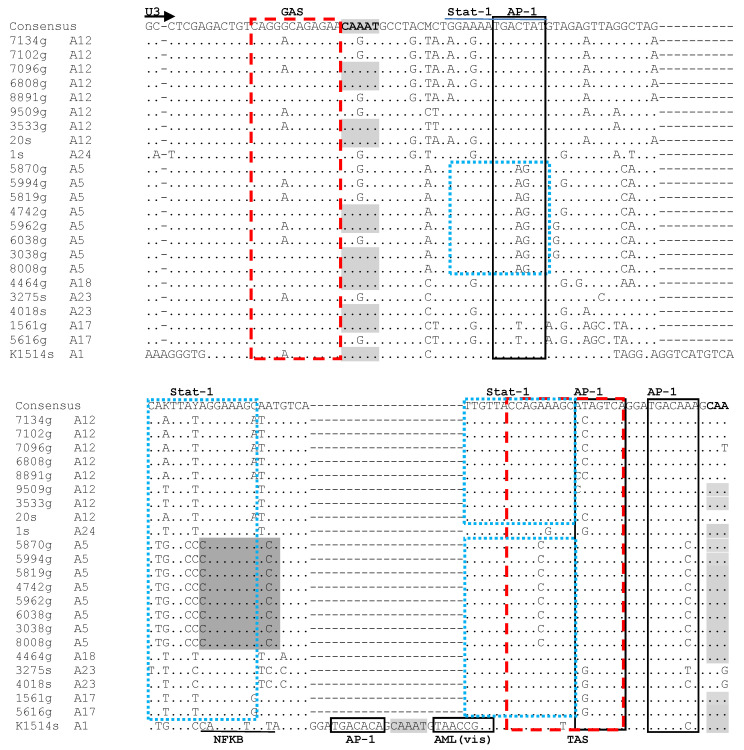
Alignment of U3-R sequences of the LTR region from Polish SRLV strains analyzed in this study. Dots indicate an identical nucleotide, and dashes represent gaps. Boundaries between U3, R and U5 are indicated by straight arrows. AP-1, AP-4, AML (vis) motifs, TATA box, GAS, TAS, Stat-1 and polyadenylation signal (poly A) are marked by boxes. NF-κB, IFR-1 and CAAT are shaded and indicated by a solid bar. s—sheep; g—goat.

**Figure 3 viruses-15-00302-f003:**
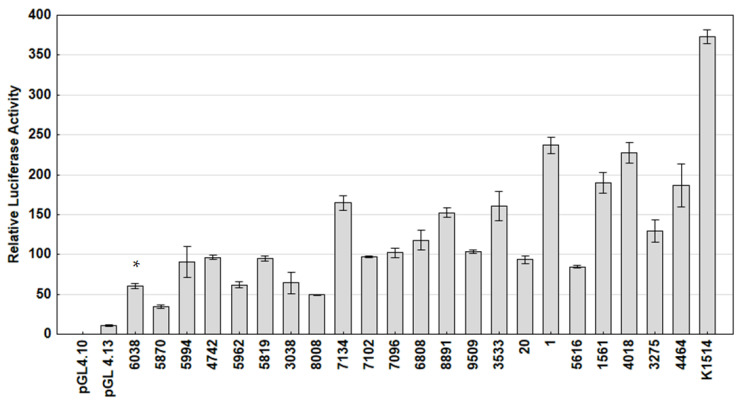
Basal transcriptional activity of SRLV LTR variants in TIGEF cells. Results present the mean of three independent transfection experiments, and standard deviations are shown as error bars. The asterisk indicates a significant difference (*p* < 0.05) with the reference strain K1514 LTR according to the Mann–Whitney U Test.

**Figure 4 viruses-15-00302-f004:**
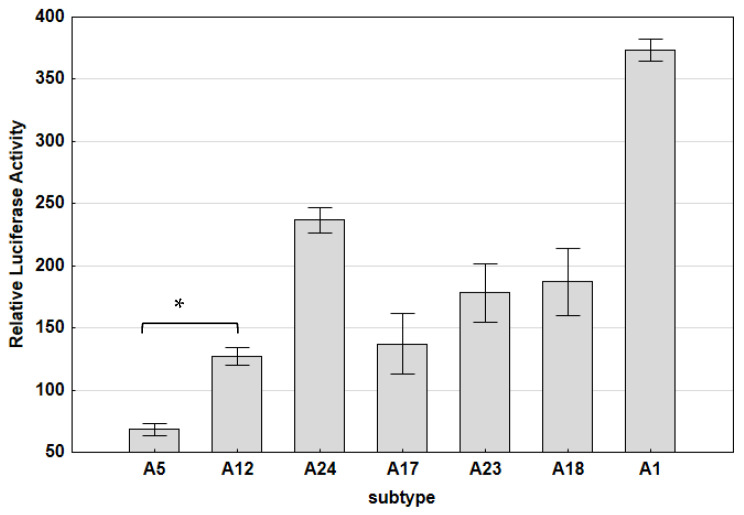
Subtype-specific transcriptional activity of Polish SRLV LTR variants in TIGEF cells. Differential activity of the SRLV LTR variants representing subtypes A1, A5, A12, A17, A18, A23 and A24. Results present the mean of independent transfection experiments, and standard deviations are shown as error bars. The asterisk indicates a significant difference (*p* < 0.05) according the Mann–Whitney U test. Subtypes A1, A17, A18, A23 and A24 were not included in the inter-subtype analysis due to limited samples (<5) representing each subtype.

## Data Availability

All data generated and analyzed in this study are included in this article.

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
