# Peer review of "Genetic Diversity of the LTR Region of Polish SRLVs and Its Impact on the Transcriptional Activity of Viral Promoters"

_viruses, 2023, doi:10.3390/v15020302_

Round 1

Reviewer 1 Report

This is an interesting ands well presented study on the promoter activity of a panel of SRLV field strains. The authors used a reporter gene assay to assess the activity of differerent SRLV LTRs. They found some differences in promoter activity between SRLV subtypes as well ass differential presence of putative transcription factor binding sites. THe results are extensively discussed and compared to that of other sudies. 

I have the follwong minor comments:

- please spell out the type of cells (TIGEF) used for transfection studies. MAybe the authors would like to discuss the impact pf the cell type used on their results. 

- Figure 3: I do not understand the rational to use the   pGL4.13 plasmid (using the SV40 promoter) as a reference for the statistical analyses. In my view it would be more solid to use one of the LTR constructs as a reference, i.e. the one of the reference strain K1214A1. In addition  it is, difficult to understand why 9509 A12 is not significantly different from the reference. while all others are. 

- Overall I think that the study could be further advanced by mechanistical investigations on the role of the different transcription factors. As it stands the interpretations of the findings remain quite speculative. 

Author Response

We would like thank the reviewer for his comments on our manuscript. We have acted upon the suggestions provided by the reviewer  and alterations were included in the updated version of the manuscript.

Reviewer 1

This is an interesting ands well presented study on the promoter activity of a panel of SRLV field strains. The authors used a reporter gene assay to assess the activity of differerent SRLV LTRs. They found some differences in promoter activity between SRLV subtypes as well ass differential presence of putative transcription factor binding sites. THe results are extensively discussed and compared to that of other sudies. 

I have the follwong minor comments:

- please spell out the type of cells (TIGEF) used for transfection studies.

Re: It has been corrected

 MAybe the authors would like to discuss the impact pf the cell type used on their results. 

Re:  It has been corrected.

- Figure 3: I do not understand the rational to use the   pGL4.13 plasmid (using the SV40 promoter) as a reference for the statistical analyses. In my view it would be more solid to use one of the LTR constructs as a reference, i.e. the one of the reference strain K1214A1. In addition  it is, difficult to understand why 9509 A12 is not significantly different from the reference. while all others are. 

Re: In response to the reviewer's comment, reference strain K1514  was used as a reference for statistical analysis. Section of the results was corrected.

- Overall I think that the study could be further advanced by mechanistical investigations on the role of the different transcription factors. As it stands the interpretations of the findings remain quite speculative. 

Re: Further study, including more strains, subtypes, cell lines as well as advanced mechanical investigations on the role of the different transcription factors will be conducted in the future. Such information is included at the end of the discussion.

Reviewer 2 Report

The authors have presented a well written manuscript that takes into account previously published data. They have analysed the promoter activity of different small ruminant lentivirus isolate LTRs, first by sequencing them and then by cloning the LTRs into a luciferase plasmid.  This was transfected into goat fibroblast cells to compare luciferase expression.  Expression was normalised using a renilla luciferase control.  

The sequencing data is presented as a phylogeny that incorporates the known lineages of SRLV and the transfection results clearly shown.  The statistics used to analyse the results are non-parametric but I wonder if the Mann-Whitney U is appropriate for groups with n=3.  Usually this should be n≥5.  The discussion is full, but the authors have not discussed possible differences that might be seen in macrophages rather than fibroblasts.  This should be incorporated as macrophages are such an important cell type in SRLV disease.

Some specific comments are below:

Line 46 ‘Only in one-third of’ delete ‘in’ to become ‘Only one-third of’

Line 58 ‘the latter way, ‘ delete ‘way’ to become ‘the latter cells, ‘

Samples – give more information on the type of disease seen in the animals from which the isolates were made.

Fig 3 – I am surprised that ‘1 A24’ is not statistically different to pGL 4.13 given the mean and sd. The use of the Mann-Whitney U test with n=3 as used in Fig 3. is inappropriate and the statistics should be changed to account for this. See my comments on the Mann-Whitney U test above. Also this is multiple comparisons so would need to correct the p value.  Changing statistical methods may change some of the conclusions given in the abstract, results and discussion.

What statistics were used in the Figure 4.

Author Response

We would like thank the reviewer for his comments on our manuscript. We have acted upon the suggestions provided by the reviewer  and alterations were included in the updated version of the manuscript.

Reviewer 2

The authors have presented a well written manuscript that takes into account previously published data. They have analysed the promoter activity of different small ruminant lentivirus isolate LTRs, first by sequencing them and then by cloning the LTRs into a luciferase plasmid.  This was transfected into goat fibroblast cells to compare luciferase expression.  Expression was normalised using a renilla luciferase control.  

The sequencing data is presented as a phylogeny that incorporates the known lineages of SRLV and the transfection results clearly shown.  The statistics used to analyse the results are non-parametric but I wonder if the Mann-Whitney U is appropriate for groups with n=3.  Usually this should be n≥5.  The discussion is full, but the authors have not discussed possible differences that might be seen in macrophages rather than fibroblasts.  This should be incorporated as macrophages are such an important cell type in SRLV disease.

 Re: In response to the reviewer's comment, Mann-Whitney test was performed only to compare promoter activity between subtype A5 (n=8) and subtype A12 (n=8) for which number of representing samples was >5. Subtypes A1, A17, A18, A23 and A24 were not included in statistical analysis due to the limited samples (<5) representing each subtype. Appropriate information was included in the text. Information about cell line was included at the end of the discussion.

Some specific comments are below:

Line 46 ‘Only in one-third of’ delete ‘in’ to become ‘Only one-third of’

Re: It has been corrected.

Line 58 ‘the latter way, ‘ delete ‘way’ to become ‘the latter cells, ‘

Re: It has been corrected.

Samples – give more information on the type of disease seen in the animals from which the isolates were made.

Re: Animals were clinically healthy, without any clinical signs and such information was included in the text.

Fig 3 – I am surprised that ‘1 A24’ is not statistically different to pGL 4.13 given the mean and sd. The use of the Mann-Whitney U test with n=3 as used in Fig 3. is inappropriate and the statistics should be changed to account for this. See my comments on the Mann-Whitney U test above. Also this is multiple comparisons so would need to correct the p value.  Changing statistical methods may change some of the conclusions given in the abstract, results and discussion.

Re: In response to the reviewer's comment, Mann-Whitney test was performed only to compare promoter activity between subtype A5 (n=8) and subtype A12 (n=8) for which number of representing samples was >5. Subtypes A1, A17, A18, A23 and A24 were not included in statistical analysis due to the limited samples (<5) representing each subtype. Appropriate information was included in the text. Furthermore, reference strain K1514  was used as a reference for statistical analysis and the results was corrected.

What statistics were used in the Figure 4.

Re: Differences in the luciferase activities between the two groups were analyzed by the Mann–Whitney U Test.